# Towards Neural Kolmogorov Equations: Parallelizable SDE Learning with Neural PDEs

Arthur Bizzi[1] and Olga Fink[1]

[1]EPFL
{arthur.coutinhobizzi, olga.fink}@epfl.ch

## Abstract

The presence of stochasticity makes learning differential equations from data substantially harder, requiring Neural SDEs to be trained with costly procedures involving repeated sequential integration. We introduce Neural Kolmogorov Equations, a parallelizable framework for learning continuous stochastic processes, based on the deterministic framework of the Forward-Kolmogorov Equation.

## 1 Introduction

Stochastic differential equations (SDEs) are a prominent framework in industry, finance, and generative modelling. We consider the time-independent Itô stochastic differential equation:

$$dX = F(X)\,dt + B(X)\,dW_t, \qquad (1)$$

where $X_t \in \mathbb{R}^d$, $F : \mathbb{R}^d \times \mathbb{R} \to \mathbb{R}^d$ is the drift, $B : \mathbb{R}^d \times \mathbb{R} \to \mathbb{R}^{d \times m}$ is the diffusion coefficient, and $W_t \in \mathbb{R}^m$ is an $m$-dimensional Wiener process.

A major challenge is learning $F$ and $B$ from samples. Current state-of-the-art Neural Stochastic Differential Equations (Neural SDEs) require costly autoregressive training [1, 2], which cannot be easily parallelized, and suffer from low accuracy, due to the limited order of stochastic integrators.

Famously, the probability distribution of the realizations of $X$ obeys the *Fokker-Planck* or *Forward-Kolmogorov* Equation (FKE):

$$\partial_t p(t,x) = L^* p = -\nabla_x \cdot \big(F(x)p\big)$$
$$+ \tfrac{1}{2}\sum_{i,j=1}^{d} \partial_{x_i}\partial_{x_j}\big(G_{ij}(x,t)\,p\big), \qquad (2)$$

where $G = BB^\top$. However, learning the FKE's coefficients is challenging for traditional methods, especially in high dimensions.

We present first the steps towards Neural Kolmogorov Equations (NKEs), a framework for learning and simulating SDEs based on the Fokker-Planck-Kolmogorov Equations. With this deterministic approach, we expect to achieve faster learning from data, via parallelizable forward-backward mixture propagation, and faster inference, by leveraging high-order deterministic numerical methods.

**Figure 1.** Propagating probabilities with NKEs.

## 2 Neural Kolmogorov Eqs.

### 2.1 Architecture

Our aim is to model the FKE as a Neural Partial Differential Equation, by learning the action of its generator $L^*$ on a basis for the ambient Hilbert space carrying our probabilities. Inspired by classical methods in filtering [3] and Lagrangian particles [4], we choose the family of Gaussian Mixtures (GMs) as our basis:

$$p(t,x) \;=\; \sum_{k=1}^{K} \pi_k\, N(x \mid \mu_k(t), \Sigma_k(t)), \qquad (3)$$

where the weights satisfy $\pi_k \geq 0$ and $\sum_{k=1}^{K} \pi_k = 1$, and each component is parametrized by a mean vector $\mu_k \in \mathbb{R}^d$ and covariance matrix $\Sigma_k \in \mathbb{R}^{d \times d}$.

The action of the generator $L^*$ on GMs is well understood: For a short time and a concentrated Gaussian, the drift $F$ carries its center $\mu$; the Jacobian $DF$ determines its stretching and rotation; the noise $G$ determines its spreading. Formally:

$$\dot{\mu}_k \approx F(\mu_k) \qquad (4a)$$
$$\dot{\Sigma}_k \;\approx\; DF\Sigma_k + \Sigma_k DF^\top + G(\mu_k) \qquad (4b)$$

We can then represent these dynamics in terms of a Neural ODE, modelling both $F$ and $G$ as $\theta$-parametrized, differentiable neural networks:

$$\dot{\mu}_k \approx F_\theta(\mu_k) \qquad (5a)$$
$$\dot{\Sigma}_k \approx DF_\theta\Sigma_k + \Sigma_k DF_\theta^\top + G_\theta(\mu_k) \qquad (5b)$$

This is the insight behind NKEs: we may now train our networks whenever there are estimates for $\dot{\mu}$ and $\dot{\Sigma}$; moreover, we now have a neural representation for $p(t,x)$ via (3) (see Fig. 1).

## 2.2 Forward-Backward Training

We now extract drift and diffusion terms from realizations of an SDE. Take a family of snapshots of realizations; we describe each snapshot in terms of a GM and evaluate the changes in mean and covariance across time steps (see Fig. 2).

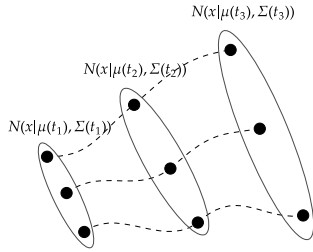

**Figure 2.** Fitting Mixtures to snapshots of a stochastic process. Our objective is to derive $F$ and $G$ from the time-derivative of $\mu$ and $\Sigma$.

One could then approximate $F$ and $G$ from two snapshots, using forward-difference approximations to eqs. (4a) and (4b). However, this approach is ill-posed for stochastic systems. To see this, consider the two snapshots in Fig. 1: Looking at the distributions alone, it is unclear if the widening of the Gaussian happened because of positive divergence of the vector field or due to the spreading effect of the noise term. Mathematically, this manifests as the problem being underdetermined; in 1-D, both $F$ and $G$ must be determined from a single equation.

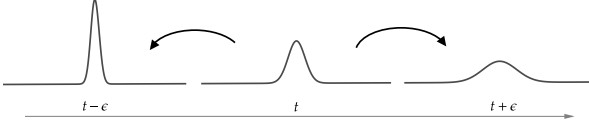

**Figure 3.** Both drift and diffusion may cause spreading.

To solve this problem, we can leverage the fundamental difference between drift and diffusion processes: time-reversibility. Vector flows are invertible and do not distinguish between past and future; diffusion, however, is entropy-increasing and thus fundamentally irreversible. Inspired by this, we can use *three* snapshots: for each time step and each mixture component, we train $F_\theta$ and $G_\theta$ so that they minimize both the forward and backward differences between the Gaussians:

$$\min_\theta \sum_n \sum_k ||\mu_k(t_{n+1}) - \mu_k(t_n) - \Delta t F_\theta(\mu_k(t_n))||_2^2$$
$$+ ||\mu_k(t_n) - \mu_k(t_{n-1}) - \Delta t F_\theta(\mu_k(t_n))||_2^2 \quad (6)$$

A similar term may be derived for the covariances $\Sigma_k$; training with the Jacobian is enabled by forward-differentiation schemes. In practice, we observe that this simple change significantly improves accuracy.

## 3 Experiment: Black-Scholes

We perform a preliminary experiment, comparing NKEs to the (classical) SDE discovery framework proposed in [5]. The stochastic system used as a metric is a minimal version of the celebrated Black-Scholes Equation, widely used in finance. For the experiment, we sample 10 trajectories at 10 equally spaced time points for the SDE:

$$dX = (f_0 + f_1 X)dt + (b_0 + b_1 X)dW_t, \quad (7)$$

where $f0 = b0 = 0$, $f_1 = 2.5$ and $b_1 = 0.4$.

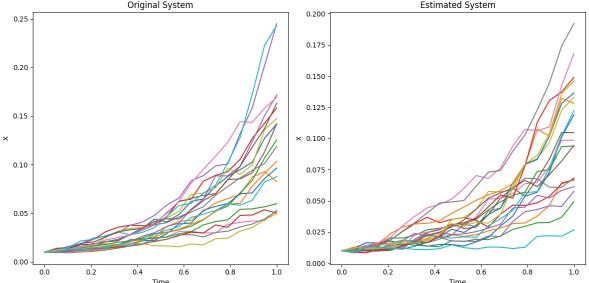

**Figure 4.** Original and estimated Black Scholes system.

For fairness, we parametrize all models as affine; we can then evaluate the accuracy of each method by analysing the coefficients obtained. The results may be found in Table 1, where NKE-fwd stands for an NKE trained using only forward differences. Samples may be visualized in Fig. 4.

**Table 1.** Parameter error for each method

| Method | $f_0$ | $f_1$ | $b_0$ | $b_1$ |
|---|---|---|---|---|
| [5] | 0.00 | 0.08 | 0.00 | 0.04 |
| NKE-fwd | 0.01 | 0.33 | 0.01 | 0.23 |
| NKE | 0.02 | 0.06 | 0.03 | 0.04 |

These results are based on early heuristics for the training and Gaussian placement; nevertheless, they remain on par with those reported in [5]. Meanwhile, our approach may be used for nonlinear, nonparametric terms $F$ and $G$. We expect improvements to the model to further increase accuracy.

## 4 Conclusion and Future Work

These first results indicate our methodology has the potential to identify SDEs from data in a parallelizable manner. Moreover, these results, along with theoretical considerations, indicate that the forward-backward scheme may indeed improve the accuracy of the learning process.

Future iterations of this work will evaluate the performance of NKEs on a broader set of benchmarks, including high-dimensional and nonlinear systems, as well as systems undergoing shocks and jumps.

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
