# OpenReview forum: "Towards Neural Kolmogorov Equations: Parallelizable SDE Learning with Neural PDEs"
_NLDL.org/2026/Abstracts_Track — NLDL 2026 Abstracts_

### Official Review · Reviewer_8PWh · 2025-10-24

**Soundness:** 4
**Correctness:** 4
**Rating:** 5
**Confidence:** 4

**Summary:**

This work proposes Neural Kolmogorov equations (NKEs) that is a framework for learning the dynamics of SDEs using neural PDEs. The authors focus on Gaussian Mixtures where the action of the Fokker-Planck equation is well understood. They argue that a forward-backward training objective is necessary for learning stochastic systems and provide a preliminary experiment on learning the dynamics of a widely used stochastic system.

**Strengths:**

This work is conceptually clean and well-formulated. The formulation of Neural Kolmogorov equations for learning SDE dynamics via neural PDEs is sound and interesting. The need for a forward-backward training objective is well-motivated and supported with empirical evidence on a simple stochastic system.

**Weaknesses:**

1. There is no comparison to neural SDE approaches which the authors note can be costly and inaccurate. It would be useful to quantify how NKEs compare in terms of speed and accuracy.
2. Experiments are limited to linear systems and affine models. As they mention, they should extend future work to high-dimensional data and nonlinear systems.
3. Future work could explore how different samplers of the stochastic system influence the learned dynamics. Or if the Gaussian Mixture assumption biased the learnt dynamics.

---

### Official Review · Reviewer_eHu4 · 2025-10-27

**Soundness:** 3
**Correctness:** 3
**Rating:** 5
**Confidence:** 4

**Summary:**

The nature of stochastic differential equations makes them challenging to learn in a similar fashion to neural ordinary differential equations. Current implementations require costly autoregressive training, which are difficult to parallelise, and the integrators are low order leading to low accuracy.

The work reformulates the learning problem to modelling the probability distributions through the Forward-Kolmogorov Equation, expressed as mixtures of Gaussians. Distinguishing between vector flows which are time reversible and noise which is not enables to use three snapshots to minimise both forward and backward differences.

The shown results are preliminary, but the work is methodological with potential for large impact on learnable stochastic differential equations. Can definitely spark discussions during the conference.

**Strengths:**

- Enabling efficient learning of stochastic differential equations with non-linear relations.
- Enabling parallelisation for learning continuous stochastic processes.
- Improved accuracy via three-point forward-backward training exploiting the different behaviour of drift and diffusion.

**Weaknesses:**

- Only demonstrated on a single system.

---

### Official Review · Reviewer_Akav · 2025-11-01

**Soundness:** 3
**Correctness:** 3
**Rating:** 4
**Confidence:** 3

**Summary:**

The abstract provides promising insights into leveraging a parallelizable Neural Kolmogorov framework for identifying stochastic differential equations and enhancing the efficiency of data-driven learning.

**Strengths:**

A very relevant topic around SDEs and the angle and parallelizable framework approach, combined with the Neural Kolmogorov framework, provides a clean, concise, and well-laid-out theoretical explanation.

**Weaknesses:**

Perhaps the authors could have discussed more about implications in different settings, but perhaps as the work progresses, it would be interesting to see the outcomes when using other benchmarks.

---

### Decision · Program_Chairs · 2025-11-05

**Decision:**

Accept

**Comment:**

The abstract is of interest to the community and should be presented at the conference.